# Music score copyright protection based on mixed low-order quaternion Franklin moments

**Qizheng Huang**[1], **Jiayi Zhu**[2]*, **Yuanjie Xian**[2‡], **Jiyou Peng**[2‡]

1 Shanghai Jincai South Seconday School, Shanghai, China, 2 School of Mechanical Engineering, Hefei University of Technology, Hefei, Anhui, China

☯ These authors contributed equally to this work.
‡ These authors also contributed equally to this work.
* 627647718@qq.com

**Data availability statement:** All relevant data are within the paper and its Supporting information files.

## Abstract

Due to the rapid growth of the digital music industry, music copyrights have become valuable intangible assets for businesses, offering exclusivity and profitability. This article takes music copyrights as an example and designs a copyright protection method for music score digital images. The zero-watermarking algorithm offers an effective and lossless means of copyright protection. Owing to their geometric invariance, orthogonal moments exhibit superior robustness, positioning them as one of the mainstream methods in the research of zero-watermarking algorithms. The current zero-watermarking algorithms based on orthogonal moments face a trade-off between robustness and discriminability. In this paper, we propose a mixed low-order moments method based on quaternion-type fractional-order moments (QTFM), which balances the global information and texture details of color image contained in QTFM. Experimental results show that the mixed low-order moments method based on QTFM exhibits superior performance in terms of robustness. In the context of using mixed low-order moment features for image analysis, Franklin moments achieve higher average structural similarity (SSIM) values than other QTFMs.

## 1 Introduction

The development of the internet has not only fueled economic growth but also introduced challenges in copyright protection in the digital age. While the online environment could have provided greater benefits to copyright holders through the broader distribution of music, any emerging phenomenon comes with dual aspects. As copyright holders enjoy the profits generated by online music, the distribution of their works has increasingly spiraled out of control, leading to more widespread infringements compared to the traditional music environment. Digital watermarking technology has emerged in recent years as an effective solution for copyright protection, addressing issues such as copyright disputes in digital images. This technique embeds copyright information directly into the original image, allowing for the

**Funding:** The author(s) received no specific funding for this work.

**Competing interests:** The authors have declared that no competing interests exist.

extraction of the embedded information later, thus ensuring the protection of digital image copyrights [1,2].

Zero-watermarking algorithms can be categorized into three main types: spatial domain zero-watermarking algorithms [3], frequency domain zero-watermarking algorithms [4], and moments domain zero-watermarking algorithms [5]. Spatial domain features are directly used to extract image characteristics. However, when geometric and image processing attacks occur, spatial domain features show significant sensitivity, regardless of whether edge or texture information is employed [6]. Features in the frequency domain lack invariance to rotation and scaling, leading to poor performance in scenarios where such transformations occur [7]. Orthogonal moments which are used to extract and represent local and global features can describe images without information redundancy compared to non-orthogonal moments [8,9]. Therefore, it is widely used in image analysis [10], pattern recognition [11], image watermarking processing [12]. Reference [13] proposes an image moment domain steganography algorithm based on orthogonal polynomials. Reference [14] introduces a face recognition method using hybrid orthogonal polynomials. Abdulhussain et al. [15] employs embedded image kernel technique and support vector machine (SVM) classification for multi-font handwritten numeral recognitinon. In addition, orthogonal moments can be extended to three-dimensional moments [16], enabling their application in the representation of 3D objects.

Quaternions have become a widely used approach in color image processing by decomposing the image into three distinct channels [17]. Examining the development of quaternion-based orthogonal moments, quaternion-type fractional moments (QTFM) can be divided into discrete and continuous moments based on the continuity of the basis function. Discrete moments include Krawtchuk moments (KM) [18], fractional Tchebyshev moments (FrTM) [19], Hahn moments (HM) [20], Dual Hahn moments (DHM) [21], Racah moments (RM) [22], and Mountain Fourier moments (MFM) [23]. Continuous orthogonal moments are grouped by coordinate system: those defined in the Cartesian coordinate system include Legendre moments (LM) [24], Gaussian Hermite moments (GHM) [25], and Chebyshev moments (CM) [26]; while those defined in the polar coordinate system include Zernike moments (ZM) [27], pseudo Zernike moments (PZM) [28], Legendre Fourier moments (LFM) [29], exponent Fourier moments (EFM) [30], log-polar Exponent-Fourier moments (LEFM) [31], polar harmonic Fourier moments (PHFM) [32], and ternary radial harmonic Fourier moments (TRHFM) [33]. Discrete moments do not involve any approximation errors, making them more suitable for high-precision image processing tasks, such as image reconstruction, compression, and denoising. In comparison, polar continuous moments demonstrate superior rotation invariance, making them more suitable for applications requiring higher degree of robustness.

The Franklin function series, notable for its unique construction and properties resembling Haar and Schauder function systems [34–37], has attracted considerable attention in mathematics. It has also been applied in signal and image processing [38,39]. The implicit construction of the classical Franklin function imposes specific constraints on its applicability. Therefore, the development of efficient and accurate computational methods for orthogonal polynomials remains a prominent focus of contemporary research. By optimizing the initial functions and partitioning the discrete Racah polynomials plane into asymmetric parts, reference [40] successfully achieves accurate recursive computation of higher-order Racah orthogonal polynomials. Asli et al. introduced a new four-term recurrence relation to compute Krawtchouk polynomials [41]. Reference [42] implements the efficient computation of Tchebichef polynomials using adaptive threshold along with x and n-directions recurrence algorithms. Mahmmod et al. [43] proposed a method for calculating Charlier polynomials

using multi-threaded parallel computation. This threading approach involves distributing independent coefficients across different threads to address performance bottlenecks.

Through the analysis of the above content, the following conclusions can be drawn: (1) Zero-watermarking algorithms based on orthogonal moments have certain advantages over other methods in term of robustness. However, when watermark image has large sizes, some existing zero-watermarking algorithms require the computation of higher-order orthogonal moments to increase the capacity of the algorithm. In practical applications, high-order moments typically correspond to local features of the image, which means they exhibit lower robustness compared to low-order moments. (2) Recursive algorithms are often effective for computing orthogonal polynomials with invariant definition intervals. However, the piecewise intervals of the Franklin polynomials change as the order of the function increases. Therefore, while general recursive algorithms have advantages in terms of calculation accuracy, it results in significant time complexity when computing the Franklin function.

The main contributions of this paper can be summarized as follows:

- The paper employs non-recursive matrix operations to implement the orthogonalization process of Franklin polynomials, avoiding the high time complexity and computational errors associated with numerical integration by directly solving for the elements.
- Construction method of mixed low-order moments feature is used to arrange and quantize QTFM, effectively enhancing the robustness and discriminability of the zero-watermarking algorithm.
- The zero-watermarking algorithm constructed using quaternion Franklin moments achieves a balance between global and local image information, enhancing the algorithm's stability against various types of image attacks.

The rest of the paper is organized as follows: Sect 2 introduces the classical Franklin system and discuss the quaternion type moments. Sect 3 proposes the accurate computation algorithm for quaternion fractional Franklin moments (QFFM). Sect 4 specifies the proposed zero-watermarking algorithm. Experiments and analysis are provided in Sect 5. Finally, Sect 6 concludes the paper.

## 2 Preliminaries

This section will introduce the definition of the classical Franklin function and presents a method for its fast and accurate computation. In addition, the quaternion type fractional-order moments (QTFM) is defined and explained.

### 2.1 Classical Franklin polynomials

Consider a sequence $\mathcal{T} = \{t_n : n \geq 0\}$ if $t_0 = 0, t_1 = 1, t_n \in (0,1)$, $\mathcal{T}$ is everywhere dense in $[0,1]$ and each point appears in $\mathcal{T}$ at most twist, it is called an admissible sequence on $[0,1]$. By means of a non-decreasing permutation, $\eta$ is obtained from $\mathcal{T}_n$.

$$\eta_n = \left\{ \tau_i^n : \tau_i^n \leq \tau_{i+1}^n, 0 \leq i \leq n-1 \right\}, \tag{1}$$

where $S_n$ denote the space of functions defined on $[0,1]$, which is left-continuous and linear on $(\tau_i^n, \tau_{i+1}^n)$ and continuous at $\tau_i^n$ if $\tau_{i-1}^n < \tau_i^n < \tau_{i+1}^n$. It is clear that $dimS_n = n+1$ and $S_{n-1} \subset S_n$, hence there exists a unique function $f_n \in S_n$ which is orthogonal to $S_{n-1}$ and $\|f_n\|_2 = 1$.

The classical Franklin system [44] $\{f_n : n \geq 0\}$ corresponding to Haar collocation points is defined as follows:

$$f_0 = 1, f_1 = \sqrt{3}\,(2x - 1).\tag{2}$$

For $n \geq 2, f_n$ is the nth Franklin function corresponding to the partition $\mathcal{T}$, where $t_n = \frac{2m-1}{2^{t+1}}$, $n = 2^t + m$, $t = 0, 1, 2, \dots, m = 1, 2, \dots, 2^t$.

## 2.2 Fast and accurate computation of Franklin polynomials

Over the years, the computational complexity of orthogonalization process for piecewise functions has limited their practical application [45]. The Franklin system, which consists of continuous piecewise linear functions, faces similar difficulties.

Considering the following linear independent groups:

$$\begin{cases} v_0 = 1, & 0 \leq x \leq 1 \\ v_1 = x, & 0 \leq x \leq 1 \\ \dots \\ v_n = \begin{cases} 0, & 0 \leq x < a_n \\ x - a_n, & a_n \leq x \leq 1 \end{cases} \end{cases},\tag{3}$$

where $a_n = \frac{\left(2n-1-2^k\right)}{2^k}$, k is the maximum integer that satisfies $2^k < 2n - 1$.

To propose a rapid computation method for the Franklin system, the non-recursive form of the orthogonalization process is given in Eq (4), where $D_0 = 1$, for $n \geq 1$, $D_n$ is the Gram determinant. Throughout the paper, the following notation is used: $v_r$ and $v_c$ denote two basis functions forming the element in the r-th row and c-th column. $a_r$ and $a_c$ represent nodes in sequence $\mathcal{T}$ corresponding to $v_r$ and $v_c$, respectively. $a_{\max}$ represents the maximum value between $a_r$ and $a_c$.

$$f_n = \frac{\Psi_n}{\sqrt{D_n D_{n-1}}}, \text{where}$$

$$\boldsymbol{D}_n = \begin{vmatrix} \langle v_1, v_1 \rangle & \langle v_2, v_1 \rangle & \dots & \langle v_n, v_1 \rangle \\ \langle v_1, v_2 \rangle & \langle v_2, v_2 \rangle & \dots & \langle v_n, v_2 \rangle \\ \vdots & \vdots & \ddots & \vdots \\ \langle v_1, v_n \rangle & \langle v_2, v_n \rangle & \dots & \langle v_n, v_n \rangle \end{vmatrix}$$

$$\Psi_n = \begin{vmatrix} \langle v_1, v_1 \rangle & \langle v_2, v_1 \rangle & \dots & \langle v_n, v_1 \rangle \\ \langle v_1, v_2 \rangle & \langle v_2, v_2 \rangle & \dots & \langle v_n, v_2 \rangle \\ \vdots & \vdots & \ddots & \vdots \\ \langle v_1, v_{n-1} \rangle & \langle v_2, v_{n-1} \rangle & \dots & \langle v_n, v_{n-1} \rangle \\ v_1 & v_2 & \dots & v_n \end{vmatrix}.\tag{4}$$

The element $d_{rc} = \langle v_r, v_c \rangle$ at any position in the matrix $\Psi_n$ and $D_n$ can be represented as:

$$d_{rc} = \begin{cases} 1 & r = 1 \wedge c = 1 \\ \frac{1}{2}\left(1 - a_{\max}^2\right) - a_{\max}(1 - a_{\max}) & r = 1 \oplus c = 1 \\ \frac{1}{3}\left(1 - a_{\max}^3\right) - \frac{1}{2}(a_r + a_c)\left(1 - a_{\max}^2\right) + a_r a_c(1 - a_{\max}) & \text{else} \end{cases},\tag{5}$$

where '$\oplus$' and '$\wedge$' represents 'Exclusive or' and 'Logical conjunction' which are logical operator. This method circumvents errors from numerical integration, especially the ill-conditioned matrices appearing in LU decomposition. Fig 1 illustrates the computation process for Franklin polynomials.

## 2.3 Quaternion type fractional-order moments

A quaternion q has one real part and three imaginary parts, which is given by [46]:

$$q = a + b\boldsymbol{i} + c\boldsymbol{j} + d\boldsymbol{k}, \tag{6}$$

where a, b, c, d are real numbers, if $a = 0$, $q$ corresponds to a pure quaternion and can be given by $\widehat{q} = b\boldsymbol{i} + c\boldsymbol{j} + d\boldsymbol{k}$.

A RGB color image $h_C(r, \theta)$ defined in polar coordinates can be represented as:

$$h_C(r, \theta) = h_R(r, \theta)\boldsymbol{i} + h_G(r, \theta)\boldsymbol{j} + h_B(r, \theta)\boldsymbol{k}, \tag{7}$$

where $h_R(r, \theta), h_G(r, \theta), h_B(r, \theta)$ are three channels of the color image respectively. To present QTFM, it is imperative to ensure that applying fractional orders to radial basis functions neither compromises their orthogonality nor incurs significant computational complexity. Let $\alpha$ be a non-zero real number, $r \in [0, 1]$, then the fractional-order radial basis functions $\{\phi_{nm}(r, \alpha), n, m \in \mathbb{Z}\}$ can be represented as follow:

$$\phi_{nm}(r, \alpha) = \sqrt{\alpha} r^{\frac{\alpha-2}{2}} \phi_{nm}(r^\alpha). \tag{8}$$

As the multiplication of quaternion does not comply with the commutative law, there are two types of QTFM with repetition m and order n:

$$\Phi_{nm}^R = \int \int_\Omega \phi_{nm}(r, \alpha) h_C(r, \theta) e^{-\mu m \theta} r dr d\theta. \tag{9}$$

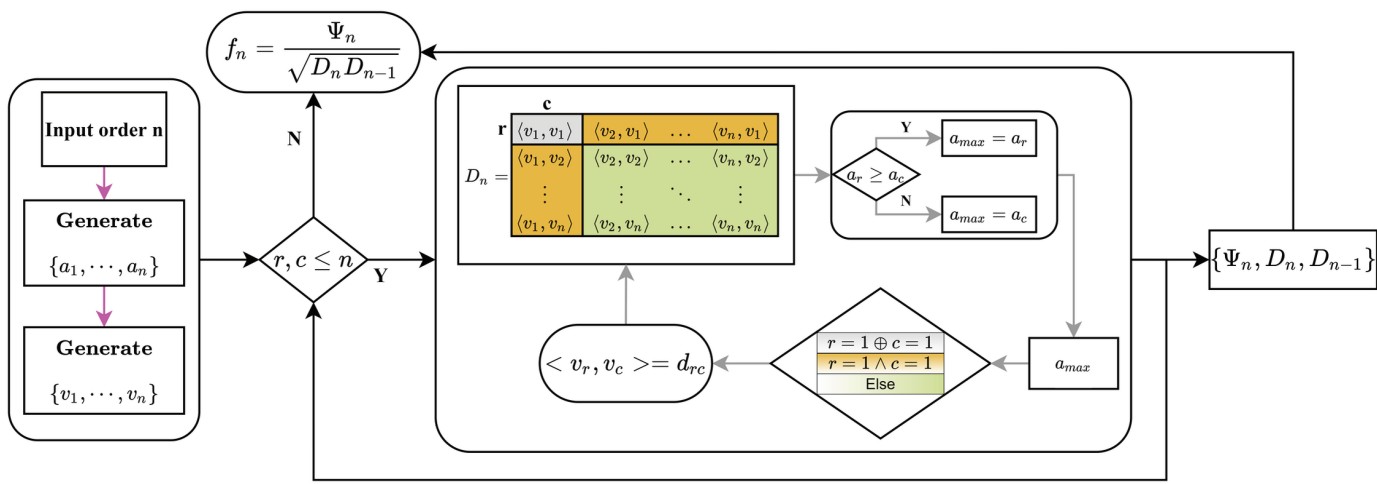

**Fig 1. Computation process for Franklin polynomials.**

$$\Phi_{nm}^L = \int \int_\Omega e^{-\mu m\theta} h_C(r,\theta)\,\phi_{nm}(r,\alpha)\,rdrd\theta, \tag{10}$$

where $\Omega$ is the definition domain of color image, $\mu$ is a pure quaternion satisfying $\|\mu\| = 1$. Due to the similar properties of $\Phi_{nm}^R$ and $\Phi_{nm}^L$, in the following chapters, we will only discuss $\Phi_{nm}^R$.

## 3 Accurate quaternion Franklin moments

In this section, the computational process of Franklin moments and necessity of precise calculation are presented. Accurate computation methods for Franklin moments are proposed using Gaussian integration and wavelet integration.

### 3.1 Definitions

For gray-level image, fractional-order Franklin moments (FFM) defined in polar coordinates is represented as:

$$\mathcal{F}_{nm} = \frac{1}{2\pi} \int_0^{2\pi} \int_0^1 h(r,\theta) f_n(r,\alpha) e^{-jm\theta} rdrd\theta, \tag{11}$$

where $h(r,\theta)$ represents a channel of $h_C(r,\theta)$, $n \in N$ is the order of $f_n(r,\alpha)$, $m \in Z$ is the repetition. The Fractional-order Franklin polynomials are orthogonal over the range $[0,1]$. Let $\hat{\theta} = \theta - \omega$, where $\omega$ denotes the rotation angle of the original image, we have:

$$\begin{aligned}
\overline{\mathcal{F}_{nm}} &= \frac{1}{2\pi} \int_0^{2\pi} \int_0^1 h(r,\theta-\omega) f_n(r,\alpha) e^{-jm\theta} rdrd\theta \\
&= e^{-jm\omega} \frac{1}{2\pi} \int_0^{2\pi} \int_0^1 h(r,\hat{\theta}) f_n(r,\alpha) e^{-jm\hat{\theta}} rdrd\hat{\theta} \\
&= \mathcal{F}_{nm} e^{-jm\omega}.
\end{aligned} \tag{12}$$

Eq (12) demonstrates rotation invariance of FFM. Assume that the image $h(x_i, y_i)$ has dimensions $[M \times N]$, the discrete form of $\mathcal{F}_{nm}$ are shown as follows:

$$\mathcal{F}_{nm} = \frac{1}{2\pi} \sum_{i=0}^{M-1} \sum_{j=0}^{N-1} h(x_i, y_i) f_n(r_{ij}, \alpha) e^{-jm\theta_{ij}}, \tag{13}$$

where $r_{ij} = \sqrt{x_i^2 + y_j^2}$, $\theta_{ij} = \arctan\left(\frac{y_i}{x_i}\right)$, QFFM can be defined as:

$$\mathcal{F}_{nm}^R = \frac{1}{2\pi} \int_0^{2\pi} \int_0^1 h_C(r,\theta) f_n(r,\alpha) e^{-\mu m\theta} rdrd\theta. \tag{14}$$

For digital images, the double integral in Eq (14) is replaced by a double summation and its discrete form of zero-order approximation is given by the following formula:

$$\mathcal{F}_{nm}^R = \frac{1}{2\pi} \sum_{i=0}^{M-1} \sum_{j=0}^{N-1} h_C(r_{ij}, \theta_{ij}) f_n(r_{ij}, \alpha) e^{-\mu m\theta_{ij}}. \tag{15}$$

In practice, the infinite series must be truncated at the finite number P. Thus, the inverse transform of Eq (14) is given by:

$$\tilde{h}_C(r,\theta) = \sum_{n=0}^{P} \sum_{m=-n}^{n} f_n(r,\alpha) \mathcal{F}_{nm}^{R} e^{\mu m \theta}, \tag{16}$$

where $\mu$ is a unit pure quaternion, for $\mu = b\boldsymbol{i} + c\boldsymbol{j} + d\boldsymbol{k}$, substituting it into Eq (14), we have:

$$\mathcal{F}_{nm}^{R}(h_C) = A_{nm}^{R} + \boldsymbol{i}B_{nm}^{R} + \boldsymbol{j}C_{nm}^{R} + \boldsymbol{k}D_{nm}^{R}, \tag{17}$$

where

$$
\begin{aligned}
A_{nm}^{R} &= -b\operatorname{Im}(\mathcal{F}_{nm}^{R}(h_R)) - c\operatorname{Im}(\mathcal{F}_{nm}^{R}(h_G)) \\
&\quad - d\operatorname{Im}(\mathcal{F}_{nm}^{R}(h_B)), \\
B_{nm}^{R} &= \operatorname{Re}(\mathcal{F}_{nm}^{R}(h_R)) + d\operatorname{Im}\left(\mathcal{F}_{nm}^{R}(h_G)\right) \\
&\quad - c\operatorname{Im}(\mathcal{F}_{nm}^{R}(h_B)), \\
C_{nm}^{R} &= \operatorname{Re}(\mathcal{F}_{nm}^{R}(h_G)) + b\operatorname{Im}\left(\mathcal{F}_{nm}^{R}(h_B)\right) \\
&\quad - d\operatorname{Im}(\mathcal{F}_{nm}^{R}(h_R)), \\
D_{nm}^{R} &= \operatorname{Re}(\mathcal{F}_{nm}^{R}(h_B)) + c\operatorname{Im}\left(\mathcal{F}_{nm}^{R}(h_R)\right) \\
&\quad - b\operatorname{Im}(\mathcal{F}_{nm}^{R}(h_G)).
\end{aligned}
\tag{18}
$$

Here, $\mathcal{F}_{nm}^{R}(h_R))$, $\mathcal{F}_{nm}^{R}(h_G))$, and $\mathcal{F}_{nm}^{R}(h_B))$ respectively represent $\mathcal{F}_{nm}$ corresponding to red, green, and blue channels. Eq (16) can be expressed as:

$$
\begin{aligned}
\tilde{h}_C(r,\theta) &= \sum_{n=0}^{P} \sum_{m=-n}^{n} \mathcal{F}_{nm}^{R} f_n(r,\alpha) e^{\mu m \theta} \\
&= \overline{h}_A(r,\theta) + \overline{h}_B(r,\theta)\boldsymbol{i} + \overline{h}_C(r,\theta)\boldsymbol{j} + \overline{h}_D(r,\theta)\boldsymbol{k},
\end{aligned}
\tag{19}
$$

where

$$
\begin{aligned}
\overline{h}_A(r,\theta) &= \operatorname{Re}(\overline{A}_{nm}) - [b\operatorname{Im}(\overline{B}_{nm}) + c\operatorname{Im}(\overline{C}_{nm}) + d\operatorname{Im}(\overline{D}_{nm})] \\
\overline{h}_B(r,\theta) &= \operatorname{Re}(\overline{B}_{nm}) + [b\operatorname{Im}(\overline{A}_{nm}) - c\operatorname{Im}(\overline{D}_{nm}) + d\operatorname{Im}(\overline{C}_{nm})] \\
\overline{h}_C(r,\theta) &= \operatorname{Re}(\overline{C}_{nm}) + [c\operatorname{Im}(\overline{A}_{nm}) + b\operatorname{Im}(\overline{D}_{nm}) - d\operatorname{Im}(\overline{B}_{nm})] \\
\overline{h}_D(r,\theta) &= \operatorname{Re}(\overline{D}_{nm}) + [d\operatorname{Im}(\overline{A}_{nm}) - b\operatorname{Im}(\overline{C}_{nm}) + c\operatorname{Im}(\overline{B}_{nm})],
\end{aligned}
\tag{20}
$$

where $\overline{h}_A(r,\theta)$ is theoretically a zero matrix, $\overline{h}_B(r,\theta)$, $\overline{h}_C(r,\theta)$, $\overline{h}_D(r,\theta)$ represent the red, green, and blue channels of the reconstructed color image, respectively. $\overline{A}_{nm}$, $\overline{B}_{nm}$, $\overline{C}_{nm}$, $\overline{D}_{nm}$ are the gray-level reconstructed image of $A_{nm}^{R}$, $B_{nm}^{R}$, $C_{nm}^{R}$, $D_{nm}^{R}$.

## 3.2 Accurate computation

As shown in Fig 2, information loss occurs during the numerical computation of orthogonal moments. To ensure accurate computation of QFFM, we employ Gaussian and wavelet integration methods.

**3.2.1 Gaussian integration.** The zeroth order approximation of double integration results in numerical integration errors. This numerical instability occurs when computing QFFM for high values of n and m. If $f(x)$ represents a 1-D function, its gaussian numerical integration

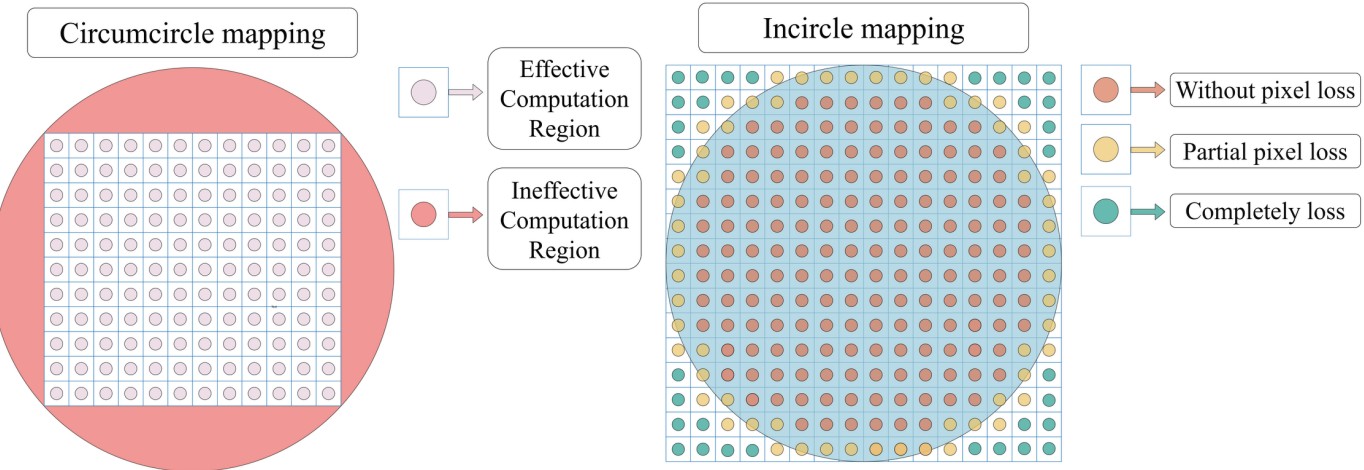

**Fig 2. Pixel information loss during the conversion of an image from Cartesian coordinates to polar coordinates.**

over the interval $[a, b]$ is expressed as:

$$\int_a^b f(x)dx \cong \frac{(b-a)}{2} \sum_{i=0}^{g-1} w_l f\left(\frac{a+b}{2} + \frac{b-a}{2}t_i\right), \tag{21}$$

where $w_i$ and $t_i$ denote the weight and position of the image sampling points, g denotes the order of Gaussian integration. The 2-D formulation of Gaussian numerical integration of $f(x, y)$ is expressed as:

$$\int_a^b \int_c^d f(x, y)\, dxdy \cong \frac{(b-a)(d-c)}{4} \sum_{l=0}^{g-1} \sum_{h=0}^{g-1} w_l w_h \times$$
$$f\left(\frac{a+b}{2} + \frac{b-a}{2}t_l, \frac{c+d}{2} + \frac{d-c}{2}t_h\right) \tag{22}$$

We achieve accurate computation of QFFM by employing Gaussian numerical integration for the double integration outlined in the following:

$$\mathcal{F}_{nm}^R = \frac{1}{MN} \sum_{l=0}^{h-1} \sum_{h=0}^{h-1} w_l w_h$$
$$\left(\sum_{i=0}^{M-1} \sum_{j=0}^{N-1} h_C\left(x_{ij}, y_{ij}\right) F_{nm}^*\left(\alpha, x_{ij}, y_{ij}\right)\right), \tag{23}$$

where $x_{ij} = \frac{t_l+2i+1-M}{M}$, $y_{ij} = \frac{t_h+2j+1-N}{N}$, $x_{ij}^2 + y_{ij}^2 \leq 1$ and $F_{nm}^*(\alpha, x_{ij}, y_{ij}) = f_n\left(r_{ij}, \alpha\right) e^{-\mu m \theta_{ij}}$.

**3.2.2 Wavelet integration.** Let $f(x)$ be an 1-D function, the wavelet numerical integration over the interval $[a, b]$ is defined as [47]:

$$\int_a^b f(x)\, dx \cong \frac{(b-a)}{2W} \sum_{l=1}^{2W} f\left(a + \frac{(b-a)(l-0.5)}{2W}\right), \tag{24}$$

where $W = 2^u$ and $u = 0, 1, 2, \ldots$, the integer u represents the level of the wavelet. The 2-D formulation of wavelet numerical integration of $f(x, y)$ is expressed as:

$$\int_a^b \int_c^d f(x, y)\, dxdy \cong \frac{(b-a)(d-c)}{4W^2} \sum_{l=1}^{2W} \sum_{h=1}^{2W} f\left(a + \frac{(b-a)(l-0.5)}{2W}, c + \frac{(d-c)(h-0.5)}{2W}\right).$$

(25)

We now accurately compute the QFFM by resorting to wavelet numerical integration of the double integration given in Eq (13):

$$\mathcal{F}_{nm}^R = \frac{1}{W^2 MN} \sum_{l=1}^{2W} \sum_{h=1}^{2W} \left( \sum_{i=0}^{M-1} \sum_{j=0}^{N-1} h_C\left(x_{ij}, y_{ij}\right) F_{nm}^*\left(\alpha, x_{ij}, y_{ij}\right) \right),$$

(26)

where $x_{ij} = \frac{2iW - MW + l - 0.5}{WM}, y_{ij} = \frac{2jW - NW + h - 0.5}{WN}, x_{ij}^2 + y_{ij}^2 \leq 1$ and $F_{nm}^*(\alpha, x_{ij}, y_{ij}) = f_n\left(r_{ij}, \alpha\right) e^{-\mu m \theta_{ij}}$.

## 4 Zero-Watermarking algorithm process

In this section, the construction method of mixed low-order moments feature and the processes of zero-watermarking algorithm are described.

### 4.1 Asymmetric tent map

Chaos theory primarily studies the behavior of dynamical systems that are highly sensitive to initial conditions. In cryptography, the sensitivity of chaotic systems to initial values is often utilized to design secure pseudo-random number generators. Asymmetric tent map [48] is defined as:

$$q_i = \begin{cases} q_{i-1}/\mu & 0 \leq q_{i-1} \leq \mu \\ (1 - q_{i-1})/(1 - \mu) & \mu < q_{i-1} \leq 1 \end{cases},$$

(27)

where $\mu \in (0, 1)$ and $q_i \in (0, 1)$ are control parameters. As shown in Fig 3, let $n = 500$ and taking the last 100 items of $q_n$, it can be observed that the values in the blue region do not exhibit a uniform distribution in the interval $[0, 1]$. In contrast, the values in the red region with $\mu \in [0.3, 0.85]$ better meet the requirements.

### 4.2 Mixed low-order moments feature

Low-order orthogonal moments represent the global features of an image, while high-order moments are more sensitive to image details. Therefore, to enhance the robustness of feature representation, mixed low-order moments method is an effective approach. Mixed low-order moments feature for any images $h_C$ can be defined as:

$$w(f) = \{|\mathcal{F}_{nm}^R| : (n, m) \in \mathbb{Z}, \alpha \in \mathbb{R}^+\},$$

(28)

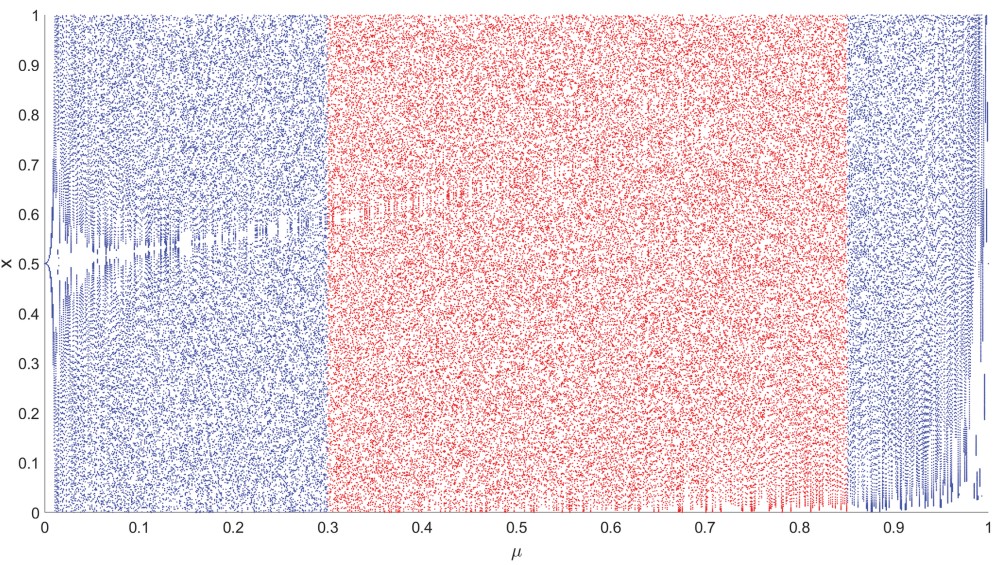

**Fig 3. The red region represents the appropriate parameter space for the asymmetric tent mapping.**

where $\mathcal{F}_{nm}^{R}$ represents moments feature given in Eq (17) and $\alpha$ is fractional-order parameter. The feature processing workflow for mixed low-order moments is shown in Fig 4.

$$bw_i = \begin{cases} 0 & |w_i| < T \\ 1 & |w_i| \geq T, \end{cases} \qquad T = \sum_{s \in \mathcal{S}} \text{median}(\boldsymbol{w}(f)) / |\alpha| \qquad (29)$$

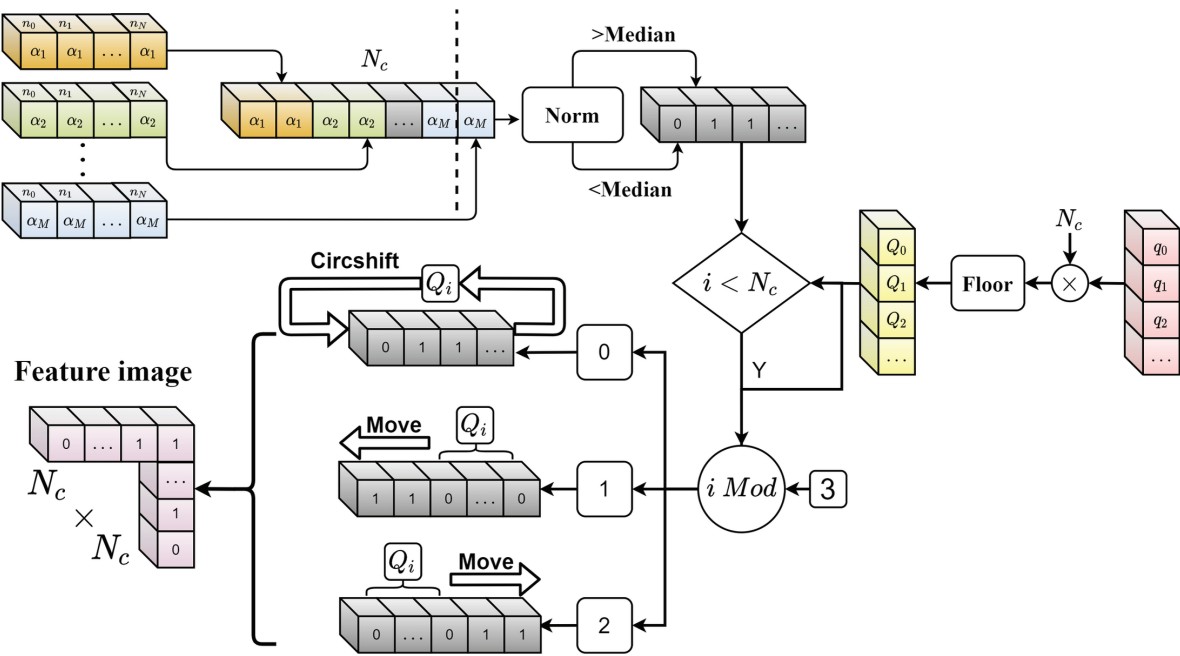

**Fig 4. Generating mixed low-Order moments feature.**

For different values of the parameter $\alpha$, the features are arranged horizontally and quantized as given in Eq (29), where $|\alpha|$ represents the number of the elements within the set $\alpha$; median($\cdot$) represents the median of a discrete sequence $w(f)$. Vector shift operations used in [49] is adopted to generate encryption feature image.

## 4.3 Zero-watermarking generation and extraction

The procedure of zero-watermarking generation is shown in Fig 5 and described as follow.

**Step 1: Applying the QTFMs on the host image.** The QTFMs is performed on the host image $h_C$ to obtain the corresponding feature $\Phi^R_{nm}$. The fractional-order parameter $|\alpha| = \{0.5, 1, 2, 3\}$.

**Step 2: Generating the Mixed low-order moments feature.**

Assuming the size of watermarking image $W_c$ is $[N_c \times N_c \times 3]$, for the QTFM with $n \in \mathbb{N}, m \in \mathbb{Z}$, the maximum order $K = \left\lceil \frac{\sqrt{2N_c+1}-3}{|\alpha|} \right\rceil$. For the QTFM with $n, m \in \mathbb{Z}$, the

maximum order $K = \left\lceil \frac{\sqrt{\frac{N_c}{|\alpha|}}-1}{2} \right\rceil$. Substituting it in Eq (28) and Eq (17), we get quantization

feature.

**STEP 3: Generating the encryption image.**

Assuming that the key1 = $\{q_0 = 0.5, \mu = 0.625, n = N_c\}$, where $n$ is iterations, $\mu$ and $q_0$ are control parameters. Then substituting key1 in the process given in Fig 4 to generate encryption image.

**STEP 4: Scrambling the encryption feature image and watermarking image.**

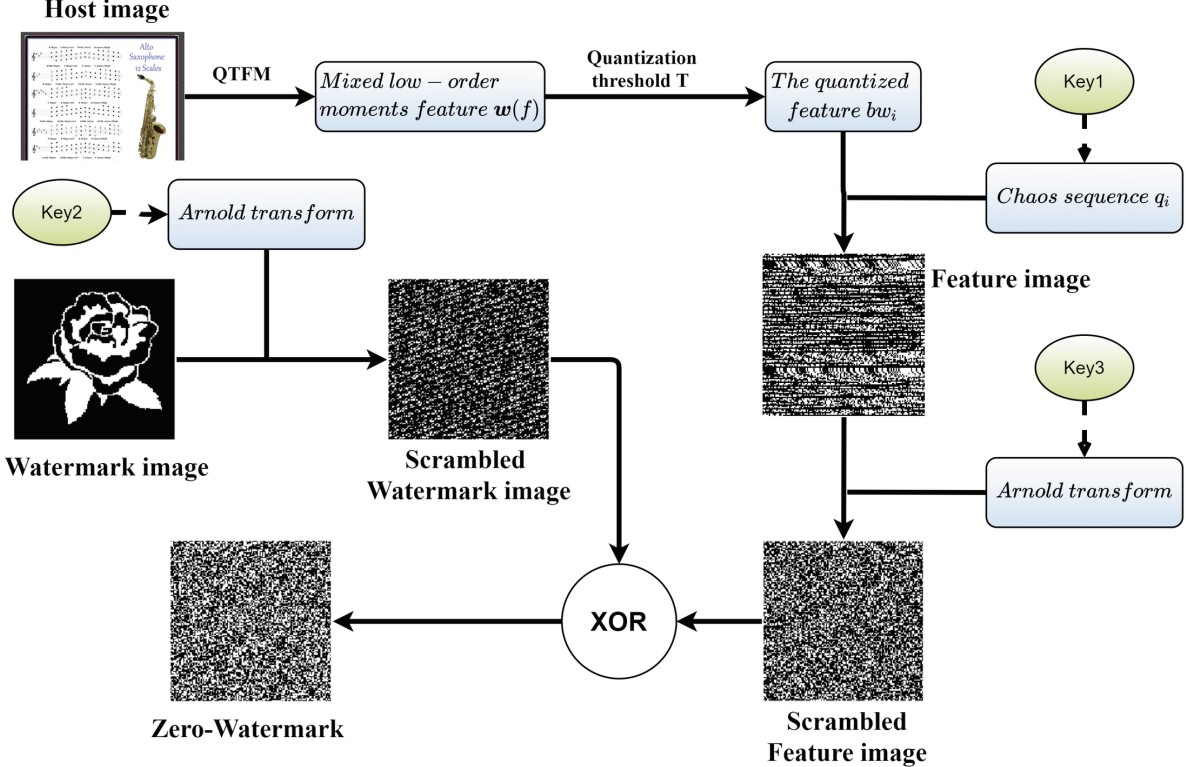

**Fig 5. Procedure of zero-watermarking generation.**

The Arnold transform is used for scrambling images which is defined as:

$$\begin{pmatrix} x' \\ y' \end{pmatrix} = \begin{pmatrix} 1 & b \\ a & ab+1 \end{pmatrix} \begin{pmatrix} x \\ y \end{pmatrix} \mod (N_c), \tag{30}$$

where $a$ and $b$ are scrambling parameters and $(x', y')$ represents the positions of the pixels $(x, y)$ after transforming. We set that key2 = $\{a = 2, b = 3, iter = 10\}$ and key3 = $\{a = 1, b = 2, iter = 5\}$, where $iter$ represents the number of iterations.

**STEP 5: Constructing the zero-watermarking.**

The zero-watermarking image is generated by performing $XOR(\cdot)$ operation which represents 'Exclusive OR'.

**STEP 6: Certifying.** The copyright owner can certify using zero-watermarking image, the key1, key2, and key3. The procedure of zero-watermarking verification is given in Fig 6. Performing the XOR of the scrambled feature image and the zero-watermarking, retrieved watermark can be obtained by inverse Arnold transform.

$$\begin{pmatrix} x' \\ y' \end{pmatrix} = \begin{pmatrix} ab+1 & -b \\ -a & 1 \end{pmatrix} \begin{pmatrix} x \\ y \end{pmatrix} \mod (N_c), \tag{31}$$

where $a, b$ is obtained from key2 and key3.

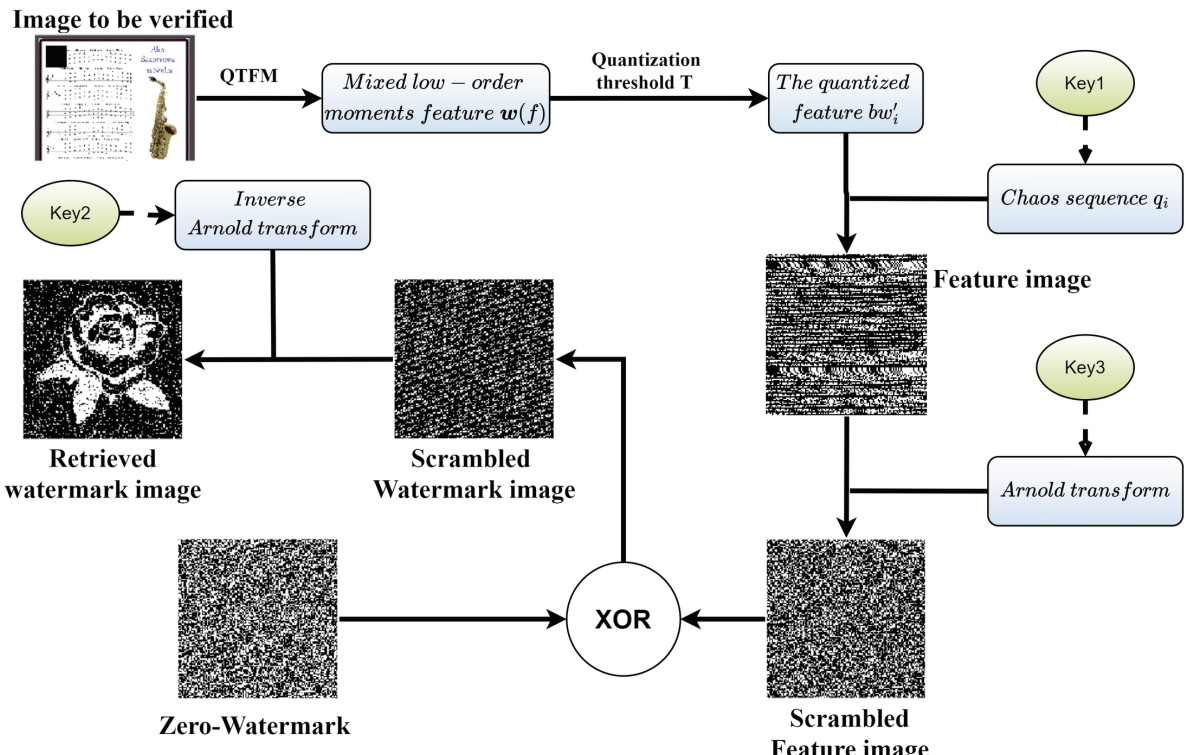

**Fig 6. Procedure of zero-watermarking verification.**

## 5 Experimental results and discussions

In this section, a set of numerical experiments are conducted to evaluate the performance of the proposed zero-watermarking algorithm and compare it with that of the existing methods for real RGB images. The dataset used in this paper is composed of both computer-generated dataset [50] and music score images collected from the internet [58] (https://imslp.org/) which is a publicly available music databases.

### 5.1 Effectiveness of the Franklin polynomial calculation method

As shown in Fig 7, the time complexity of recursive method, non-recursive method, and the proposed method are compared.

The classical Gram-Schmidt process has a time complexity of $O(CR^2)$ [51], where R is the number of column vectors and C is the dimensionality of the vectors. The time complexity of recursive orthogonalization process is $O(kn^5)$, depending on the number of nodes in sequence $\mathcal{T}$. The time complexity of the non-recursive form in Eq (4) is approximately $O(Dn^3)$, where $D$ represents the grid density for numerical integration. Since all elements in the matrix correspond to the partition $\eta_2$, the time complexity of proposed method given in Eq (5) is $O(3n^3)$.

### 5.2 Image reconstruction

Using higher-order moments for image reconstruction is an effective method for assessing the accuracy of quaternion moments, where the reconstructed image can be evaluated using the mean squared reconstruction error (MSRE) [52]:

$$\text{MSRE} = \frac{\sum \iint_{x^2+y^2<1} [h_C(r,\theta) - \tilde{h}_C(r,\theta)]^2 dxdy}{\sum \iint_{x^2+y^2<1} [h_C(r,\theta)]^2 dxdy}, \tag{32}$$

where $h_C(r,\theta)$ represents original color image and $\tilde{h}_C(r,\theta)$ represents reconstructed color image.

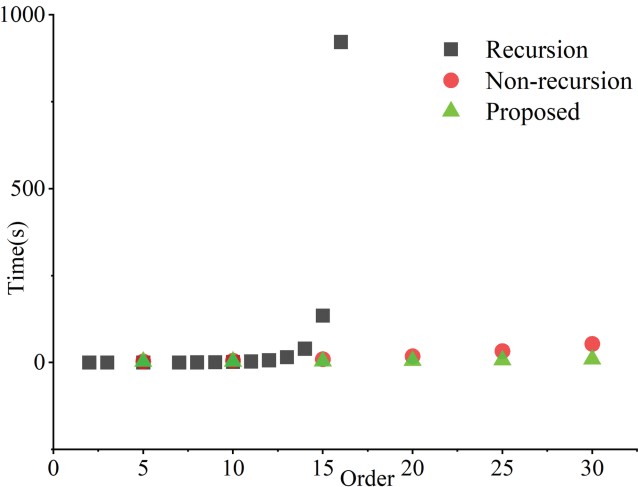

**Fig 7. The time complexity of recursive method, non-recursive method, and the proposed method.**

The proposed QFFM, FLFM, and FPCET are used to reconstruct the "Baboon" color image of size 128×128 using both direct and accurate computation methods. The computed results of MSRE are displayed in Fig 8. As given in the comparison in Fig 9, when the maximum order is less than 20, the direct computation method does not introduce significant computational errors compared to the exact computation method. However, when the maximum order exceeds 30, the direct computation method are instable. In contrast, the Gaussian integration, wavelet integration, and FFT-based methods are more suitable for tasks requiring high-precision calculation. Fig 10 illustrates the comparison of time complexity between the direct computation method and the exact computation method. Since the Gaussian integration and wavelet integration methods directly increase the grid density of the discrete numerical integration, their time complexity increases significantly. Therefore, the direct computation method is used in the zero-watermarking algorithm to obtain the mixed low-order moments.

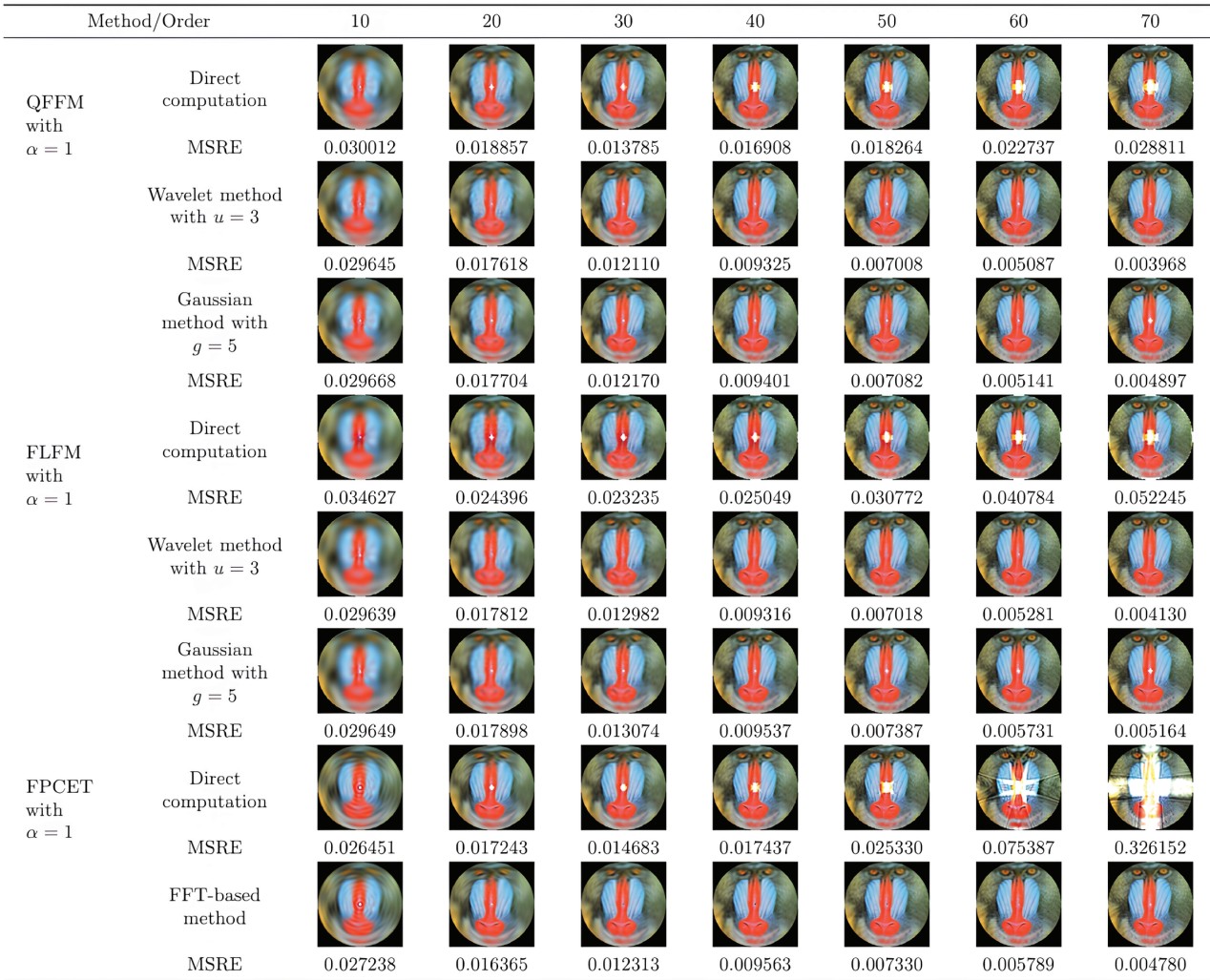

| Method/Order | | 10 | 20 | 30 | 40 | 50 | 60 | 70 |
|---|---|---|---|---|---|---|---|---|
| QFFM with $\alpha = 1$ | Direct computation | | | | | | | |
| | MSRE | 0.030012 | 0.018857 | 0.013785 | 0.016908 | 0.018264 | 0.022737 | 0.028811 |
| | Wavelet method with $u = 3$ | | | | | | | |
| | MSRE | 0.029645 | 0.017618 | 0.012110 | 0.009325 | 0.007008 | 0.005087 | 0.003968 |
| | Gaussian method with $g = 5$ | | | | | | | |
| | MSRE | 0.029668 | 0.017704 | 0.012170 | 0.009401 | 0.007082 | 0.005141 | 0.004897 |
| FLFM with $\alpha = 1$ | Direct computation | | | | | | | |
| | MSRE | 0.034627 | 0.024396 | 0.023235 | 0.025049 | 0.030772 | 0.040784 | 0.052245 |
| | Wavelet method with $u = 3$ | | | | | | | |
| | MSRE | 0.029639 | 0.017812 | 0.012982 | 0.009316 | 0.007018 | 0.005281 | 0.004130 |
| | Gaussian method with $g = 5$ | | | | | | | |
| | MSRE | 0.029649 | 0.017898 | 0.013074 | 0.009537 | 0.007387 | 0.005731 | 0.005164 |
| FPCET with $\alpha = 1$ | Direct computation | | | | | | | |
| | MSRE | 0.026451 | 0.017243 | 0.014683 | 0.017437 | 0.025330 | 0.075387 | 0.326152 |
| | FFT-based method | | | | | | | |
| | MSRE | 0.027238 | 0.016365 | 0.012313 | 0.009563 | 0.007330 | 0.005789 | 0.004780 |

**Fig 8. Reconstructed image of 'Baboon' of size 128 × 128 using FFT-based method, Gaussian numerical integration and wavelet numerical integration method.**

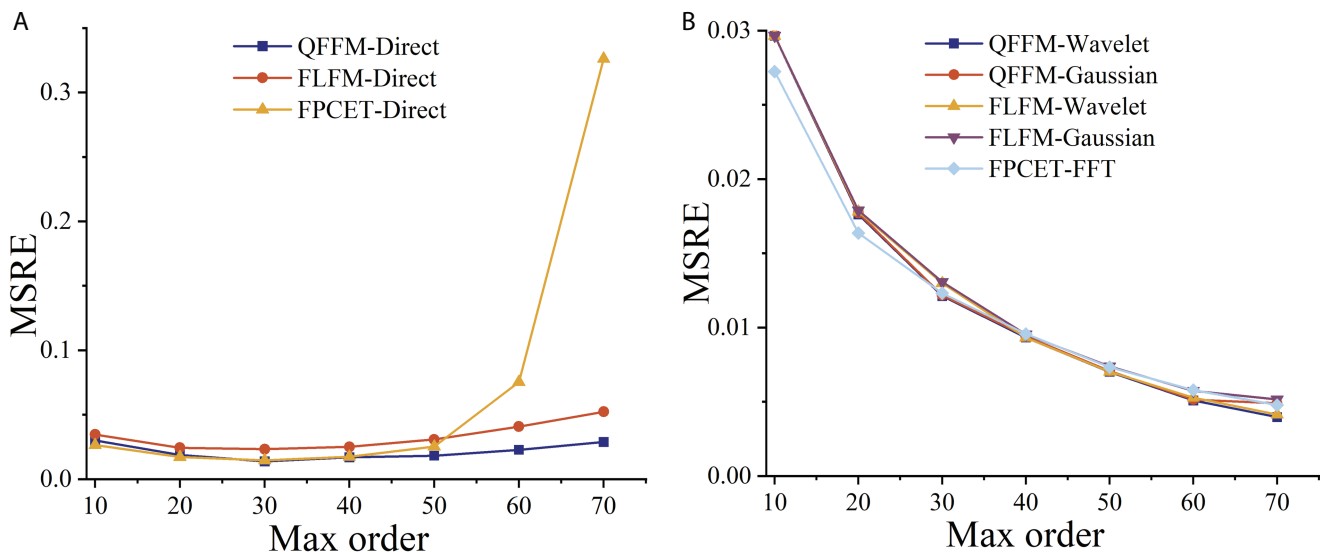

**Fig 9. Comparative study of different QTFM-based methods with Gaussian integration, wavelet integration, direct computation and FFT-based methods.**

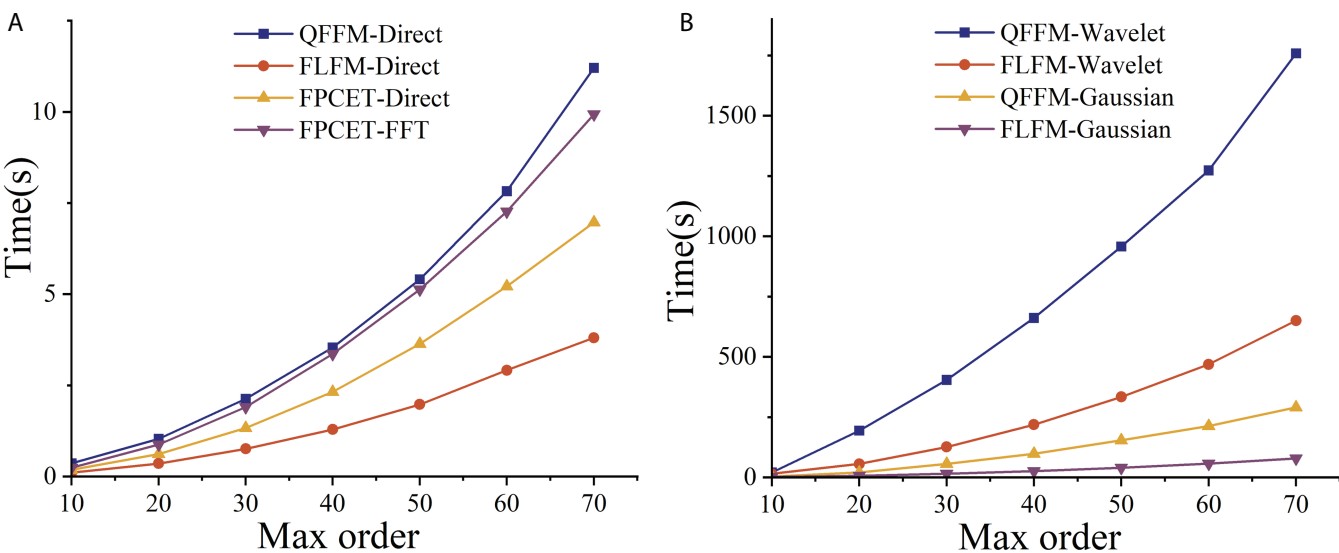

**Fig 10. The processing time of different QTFM methods based on Gaussian integration, wavelet integration, direct computation, and FFT.**

## 5.3 Robustness of proposed QFFM

Since the zero-watermarking algorithm does not affect the original image, imperceptibility is not a concern; only robustness needs to be considered. Structural similarity (SSIM) [53] used to evaluate the robustness of algorithm is defined as:

$$\text{SSIM}(x,y) = [l(x,y)]^{\alpha} \cdot [c(x,y)]^{\beta} \cdot [s(x,y)]^{\beta}$$

where

$$
\begin{aligned}
l(x,y) &= \frac{2\mu_x\mu_y + C_1}{\mu_x^2 + \mu_y^2 + C_1}, \\
c(x,y) &= \frac{2\sigma_x\sigma_y + C_2}{\sigma_x^2 + \sigma_y^2 + C_2}, \\
s(x,y) &= \frac{\sigma_{xy} + C_3}{\sigma_x\sigma_y + C_3}
\end{aligned}
\qquad (33)
$$

where $\mu_x, \mu_y, \sigma_x, \sigma_y$, and $\sigma_{xy}$ are the local means, standard deviations, and cross-covariance for images $x, y$. The value closer to 1 indicates better image quality. The similarity between the original watermark image and the extracted watermark image is evaluated using SSIM as the assessment metric. The Peak Signal-to-Noise Ratio (PSNR) [54] can be used to assess the distortion level of the attacked image, which is defined as:

$$\text{PSNR} = 10 \times \log_{10} \frac{255^2}{\text{MSE}}, \qquad (34)$$

where MSE denotes the mean square error:

$$\text{MSE} = \frac{1}{MN} \sum_{x=1}^{M} \sum_{y=1}^{N} [h(x,y) - h^*(x,y)]^2, \qquad (35)$$

where $h(x,y)$ and $h^*(x,y)$ represent the original image and attacked image, respectively, with dimensions of $M \times N$. The higher PSNR value, the lower distortion degree.

In this section, the robustness of the proposed algorithm is evaluated for the common geometric attacks, image processing attacks, and mixed image attacks. In the conducted experiments , the "saxophone" image of size $1500 \times 1200$ is selected from the collected musical score images as an example. The "flower" image of size $128 \times 128$ from Fig 11 is selected as the watermark image. Fig 12 shows the parameter settings for each attack, the PSNR values of the attacked images, and the SSIM values of the watermark images retrieved using the proposed algorithm. The obtained results shows that the SSIM values of the proposed algorithm are all greater than 0.99, indicating that the retrieved watermarks are still recognizable despite the original color image being seriously distorted.

The seven color images in Fig 11 are subjected to single-type image attacks with various parameters. Table 1 provides a summary of the robustness of the proposed algorithm against these image attacks. Table 2 summarizes the robustness of the algorithm against single-type image attacks and mixed image attacks, with the specific parameters provided in Table 3. Table 2 calculates the average test results for 50 collected music score images. The results show that the SSIM values for the proposed algorithm are greater than 0.99 for single-type image attacks and exceed 0.98 for mixed image attacks.

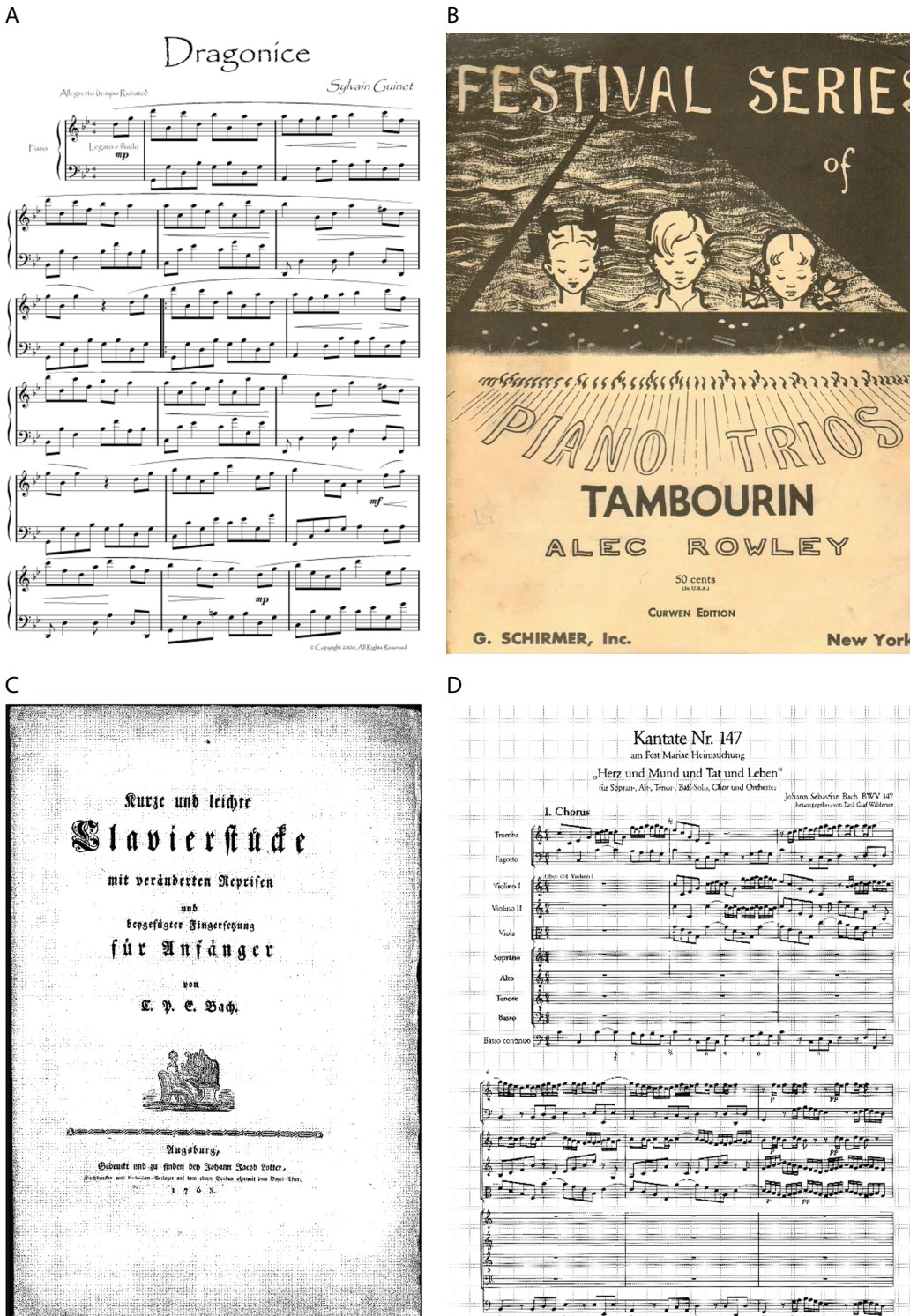

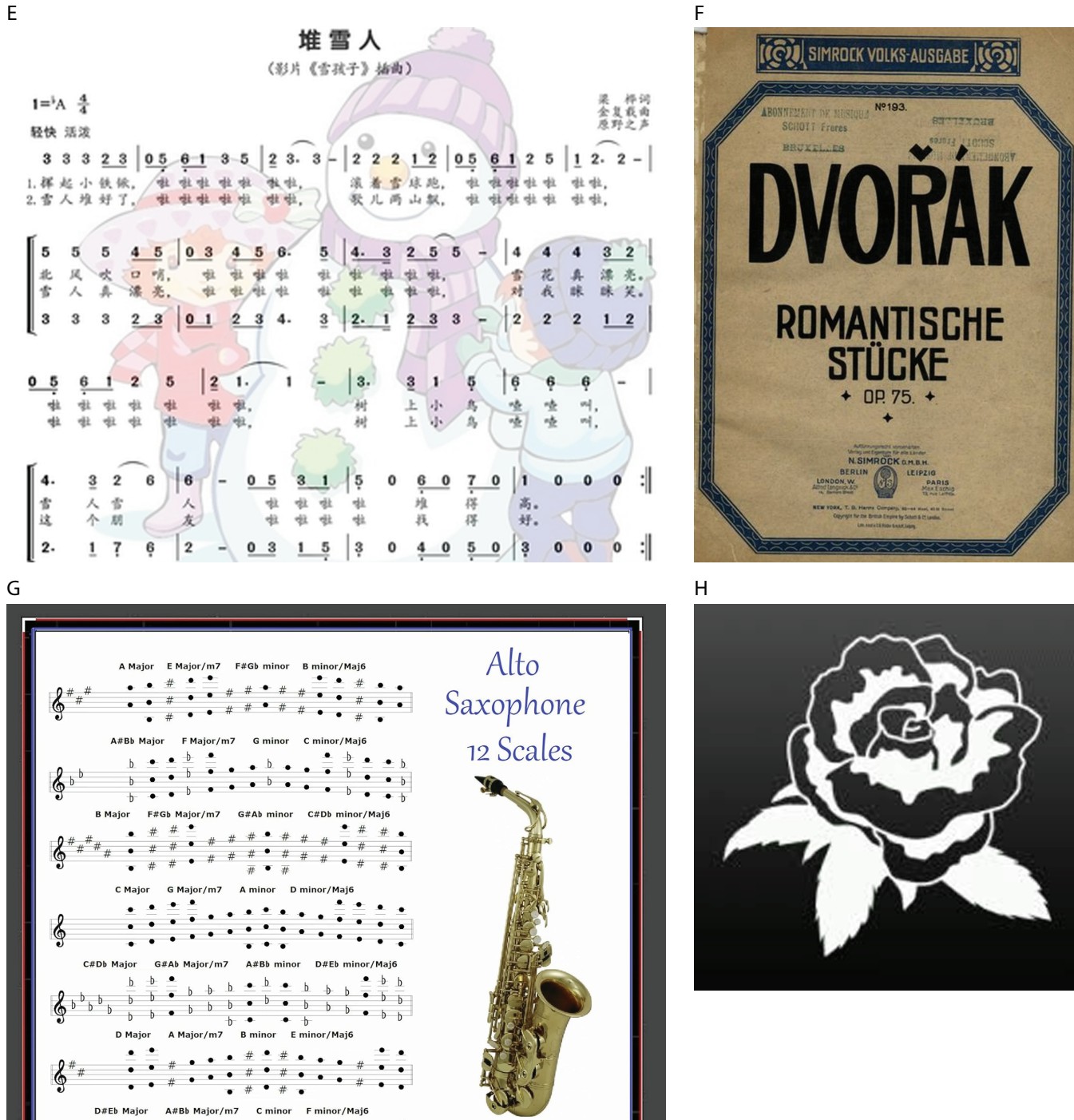

**Fig 11. Original 7 images and Logo image.** a) 'Dragonice' (b) 'Tambourin' (c) 'Klavierstucke' (d) 'Fest' (e) 'Snow child' (f) 'Dvorak' (g) 'Saxophone' (h) Watermark image.

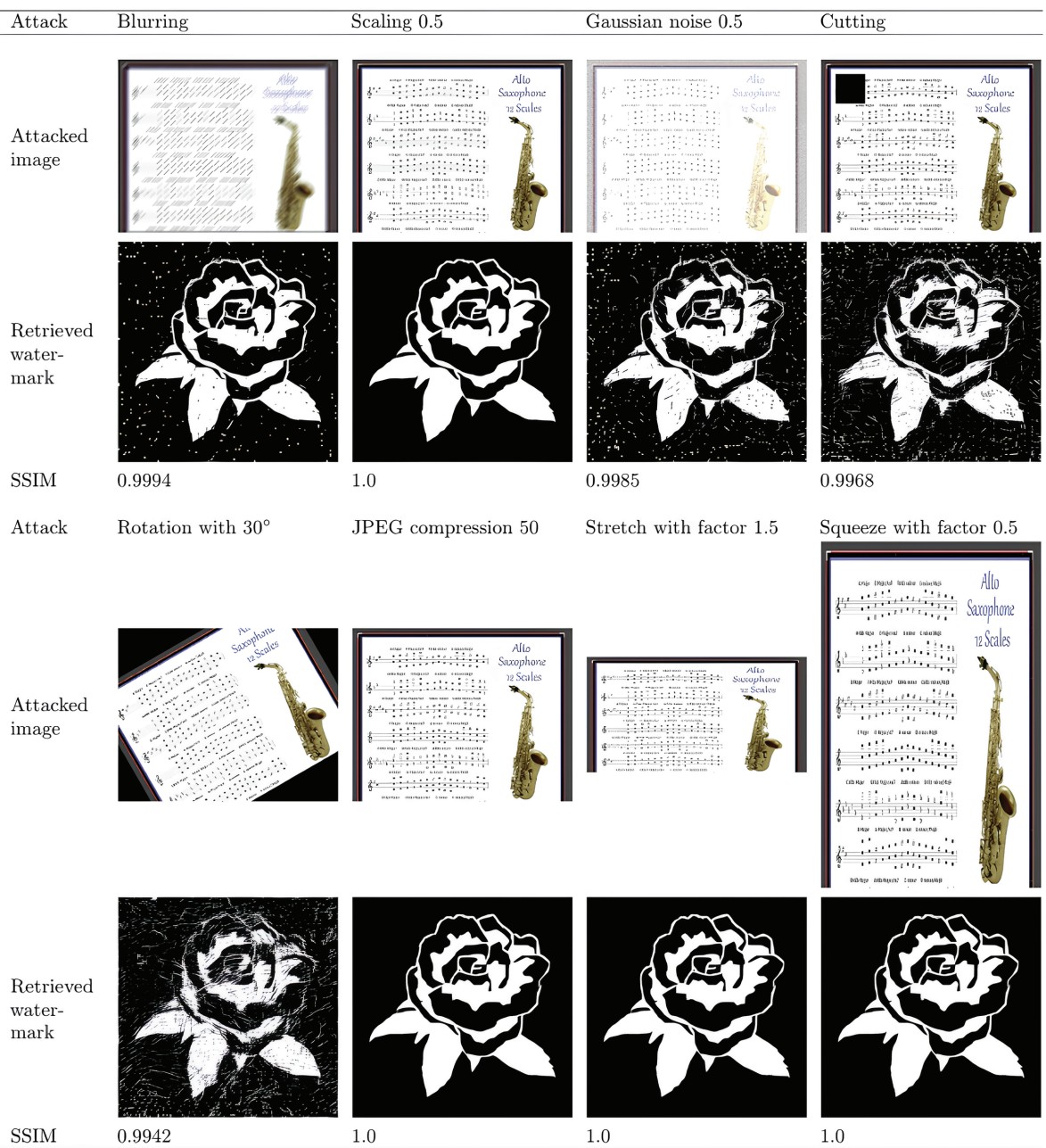

**Fig 12. Attacked images and the verified watermark image under: Blurring with PSNR = 16.8789, Gaussian noise 0.5 with PSNR = 13.8986, Cutting with PSNR = 17.9834, Rotation** $30°$ **with PSNR = 8.1672, JPEG compression 0.5 with PSNR = 39.3218.**

## 5.4 Comparison with other methods

To obtain more reliable experimental results, Table 3 compares the proposed method with the existing similar zero-watermarking algorithms [55–57] and other QTFMs. The average SSIM values are calculated using 30 test color images. The results demonstrate the robustness of the proposed method, which performs well against various types of image attacks and mixed

**Table 1. SSIM values of 7 images under common attacks with different parameters.**

| Attack | Parameters | 'Dragnonice' | 'Snow Child' | 'Tambourin' | 'Fest' | 'Klavierstucke' | 'Dvorak' | 'Saxophone' |
|---|---|---|---|---|---|---|---|---|
| Blurring | 'motion' with $\{15,15\}$ | 0.9981 | 0.9991 | 1 | 0.9988 | 0.9997 | 0.9997 | 1 |
| Blurring | 'motion' with $\{50,25\}$ | 0.9942 | 0.9982 | 1 | 0.9980 | 0.9973 | 0.9997 | 1 |
| Blurring | 'motion' with $\{25,45\}$ | 0.9979 | 0.9991 | 1 | 0.9991 | 0.9992 | 1 | 0.9999 |
| Scaling | 1.25 | 0.9835 | 0.9991 | 1 | 0.9995 | 0.9997 | 1 | 0.9992 |
| Scaling | 1.5 | 1 | 0.9991 | 1 | 0.9995 | 0.9994 | 1 | 1 |
| Scaling | 0.25 | 0.9997 | 0.9839 | 1 | 0.9997 | 0.9977 | 1 | 0.9874 |
| Cutting | $390 \times 390$ | 0.9903 | 0.9950 | 0.9997 | 0.9924 | 0.9926 | 0.9948 | 0.9950 |
| Cutting | $290 \times 290$ | 0.9904 | 0.9920 | 0.9995 | 0.9899 | 0.9852 | 0.9908 | 0.9939 |
| Cutting | $90 \times 90$ | 0.9927 | 0.9956 | 1 | 0.9936 | 0.9934 | 0.9948 | 0.9998 |
| Gaussian noise | 0.2 | 0.9992 | 0.9979 | 0.9997 | 1 | 0.9985 | 1 | 0.9991 |
| Gaussian noise | 0.25 | 0.9988 | 0.9975 | 0.9997 | 0.9997 | 0.9985 | 1 | 0.9991 |
| Gaussian noise | 0.1 | 0.9994 | 0.9985 | 0.9997 | 1 | 0.9995 | 1 | 0.9995 |
| Rotation | 5 | 0.9910 | 0.9968 | 0.9999 | 0.9951 | 0.9966 | 0.9992 | 0.9975 |
| Rotation | 15 | 0.9864 | 0.9958 | 0.9962 | 0.9924 | 0.9950 | 0.9961 | 0.9961 |
| Rotation | 45 | 0.9848 | 0.9898 | 0.9856 | 0.9889 | 0.9916 | 0.9890 | 0.9887 |
| JPEG | 10 | 0.9994 | 0.9999 | 1 | 0.9995 | 0.9991 | 1 | 1 |
| JPEG | 30 | 0.9997 | 0.9996 | 1 | 0.9997 | 0.9988 | 1 | 1 |
| JPEG | 70 | 1 | 1 | 1 | 1 | 0.9992 | 1 | 1 |

**Table 2. SSIM values of 7 images and the mean values of 50 images under mixed attacks.**

| Attack | 'Dragnonice' | 'Snow Child' | 'Tambourin' | 'Fest' | 'Klavierstucke' | 'Dvorak' | 'Saxophone' | Average |
|---|---|---|---|---|---|---|---|---|
| Blurring | 0.9990 | 0.9988 | 1 | 0.9984 | 0.9981 | 0.9999 | 0.9994 | 0.9992 |
| Scaling | 1 | 0.9839 | 1 | 1 | 0.9994 | 1 | 1 | 0.9993 |
| Gaussian noise | 0.9971 | 0.9963 | 0.9997 | 0.9988 | 0.9959 | 0.9997 | 0.9985 | 0.9982 |
| Cutting | 0.9896 | 0.9933 | 0.9995 | 0.9924 | 0.9920 | 0.9948 | 0.9968 | 0.9936 |
| Rotation | 0.9885 | 0.9924 | 0.9911 | 0.9899 | 0.9929 | 0.9896 | 0.9942 | 0.9903 |
| JPEG | 0.9997 | 1 | 1 | 1 | 0.9992 | 1 | 1 | 0.9998 |
| Stretch | 1 | 0.9839 | 1 | 1 | 0.9997 | 1 | 1 | 0.9972 |
| Squeeze | 1 | 0.9839 | 1 | 1 | 0.9997 | 1 | 1 | 0.9973 |
| Blurring+Gaussian noise | 0.9896 | 0.9953 | 0.9996 | 0.9907 | 0.9934 | 0.9991 | 0.9979 | 0.9926 |
| Blurring+Cutting | 0.9903 | 0.9933 | 0.9995 | 0.9924 | 0.9920 | 0.9936 | 0.9963 | 0.9934 |
| Blurring+Rotation | 0.9900 | 0.9917 | 0.9904 | 0.9891 | 0.9929 | 0.9888 | 0.9936 | 0.9905 |
| Gaussian noise+Cutting | 0.9947 | 0.9926 | 0.9941 | 0.9924 | 0.9929 | 0.9933 | 0.9947 | 0.9931 |
| Gaussian noise+Rotation | 0.9927 | 0.9923 | 0.9873 | 0.9892 | 0.9935 | 0.9899 | 0.9927 | 0.9891 |
| Cutting+Rotation | 0.9835 | 0.9893 | 0.9809 | 0.9899 | 0.9872 | 0.9886 | 0.9921 | 0.9863 |

**Table 3. Comparative study of average SSIM values between the proposed method and similar zero-watermarking algorithms.**

| Attack | Parameters | Proposed method | QLFM | QRHFM | QCHFM | [55] | [56] | [57] |
|---|---|---|---|---|---|---|---|---|
| Blurring | $\{50,45\}$ | 0.9994 | 0.9997 | 0.9993 | 0.9995 | 0.9968 | 0.9921 | 0.9945 |
| Scaling | 0.5 | 0.9996 | 0.9987 | 0.9997 | 0.9993 | 0.9979 | 0.9981 | 0.9990 |
| Gaussian noise | 0.5 | 0.9985 | 0.9977 | 0.9990 | 0.9988 | 0.9947 | 0.9926 | 0.9962 |
| Cutting | $200 \times 200$ | 0.9968 | 0.9972 | 0.9965 | 0.9963 | 0.9927 | 0.9928 | 0.9915 |
| Rotation | $30°$ | 0.9942 | 0.9935 | 0.9929 | 0.9944 | 0.9924 | 0.9919 | 0.9916 |
| JPEG | 50 | 0.9999 | 0.9999 | 0.9998 | 0.9999 | 0.9996 | 0.9948 | 0.9994 |
| Stretch | 1.5 | 0.9989 | 0.9991 | 0.9988 | 0.9989 | 0.9982 | 0.9983 | 0.9979 |
| Squeeze | 0.5 | 0.9991 | 0.9990 | 0.9991 | 0.9989 | 0.9981 | 0.9978 | 0.9988 |
| Blurring+Gaussian noise | $\{50,45\}, 0.5$ | 0.9979 | 0.9974 | 0.9988 | 0.9985 | 0.9933 | 0.9912 | 0.9928 |
| Blurring+Cutting | $\{50,45\}, 200 \times 200$ | 0.9963 | 0.9981 | 0.9959 | 0.9960 | 0.9917 | 0.9908 | 0.9936 |
| Blurring+Rotation | $\{50,45\}, 30°$ | 0.9941 | 0.9931 | 0.9926 | 0.9947 | 0.9907 | 0.9909 | 0.9901 |
| Gaussian noise+Cutting | $0.5, 200 \times 200$ | 0.9956 | 0.9961 | 0.9955 | 0.9973 | 0.9916 | 0.9911 | 0.9921 |
| Gaussian noise+Rotation | $0.5, 30°$ | 0.9929 | 0.9923 | 0.9939 | 0.9927 | 0.9913 | 0.9905 | 0.9907 |
| Cutting+Rotation | $200 \times 200, 30°$ | 0.9931 | 0.9935 | 0.9932 | 0.9902 | 0.9881 | 0.9907 | 0.9903 |
| Average | | 0.99688 | 0.99681 | 0.99679 | 0.99681 | 0.99408 | 0.99313 | 0.99418 |

image attacks. When using the mixed low-order moments method, the average SSIM value increases around $7 \times 10^{-5}$ compared to other types of orthogonal moments. Compared to references [55–57], the improvement is around $2.7 \times 10^{-3}$. The stability of the experimental results can be attributed to the following factors: (1) The mixed low-order moments method is used to extract low-frequency information from color images, making it is robust to image attacks that alter pixel values, such as noise, blurring, and cutting. (2) References [56,57] embed watermarking image information into the original image by altering the moments amplitudes. This method makes the algorithm sensitive to variations in image pixel values. The method employed in reference [55] requires the computation of high-order moments when the embedded watermark image has a large size, which increases the time complexity and results in relatively poor robustness. (3) The QFFM demonstrates a smaller MSRE in image reconstruction experience, indicating that, at the same maximum order, the QFFM method retains more image texture information. This enhances its robustness against geometric transformations, including scaling, stretching, and compression. (4) Excessive image texture information will lead to increased sensitivity of the algorithm to local pixel changes. Therefore, combining QFFM with the mixed low-order moments method achieves a balance between global and local image information, enhancing the stability of the proposed algorithm against various types of image attacks.

Although the proposed method has several advantages, it also has some limitations. Compared to other QTFM methods that also utilize mixed low-order moments, Franklin moments demonstrate superior average SSIM values in resisting various types of image attacks. However, for specific types of image attacks, it is possible to identify better QTFM methods, indicating that Franklin moments lack specificity in certain scenarios. Additionally, since Franklin functions are composed of piecewise linear functions, they exhibit higher time complexity as shown in Fig 10.

## 6 Conclusion

A zero-watermarking algorithm based on mixed low-order Franklin moments is proposed for to address issue of copyright protection for digital music score images. The algorithm is constructed based on a set of fractional-order parameters, which are varied within a specific range. Subsequently, a quantization scheme is applied to the extracted features to generate a binary sequence for watermark generation and extraction. The algorithm combines the ability of mixed low-order moments method to effectively extract global image features with the capacity of Franklin moments to capture detailed local information in color images, thereby enhancing its robustness against various types of image attacks. Numerical experiments obviously show that our proposal is much more robust compared to the other considered methods and QTFMs. These results clearly demonstrate that our proposed method can be effectively against common image processing operations and mixed image attacks.

Although the proposed fast computation approach for Franklin functions has achieved certain success, numerical instability issues persist when dealing with high-order (order > 200) Franklin polynomials. Therefore, developing a numerically stable construction of Franklin functions is a crucial direction for future work. Furthermore, the derivatives of classical Franklin functions demonstrate discontinuities at the boundaries of the segments. Consequently, the development of a smoother generalization of the Franklin functions represents a significant objective for future research.

## Supporting information

**S1 Dataset. Music score dataset is used in this paper.** We obtained the music score images from publicly available music databases, specifically from the IMSLP (International Music Score Library Project). This platform offers a vast collection of music scores that are in the public domain, as well as some modern works with explicit permission. The music score images can be accessed through the following link: https://imslp.org/

The majority of the music score images are sourced from the public domain, and therefore do not have any copyright restrictions. Specifically, the scores selected in our study belong to works in the public domain on the IMSLP platform, which are not protected by current copyright laws.
(ZIP)

## Author contributions

**Conceptualization:** Qizheng Huang.

**Data curation:** Jiayi Zhu, Yuanjie Xian.

**Formal analysis:** Yuanjie Xian, Jiyou Peng.

**Investigation:** Kang Huang, Jiyou Peng.

**Methodology:** Jiayi Zhu.

**Project administration:** Kang Huang.

**Resources:** Qizheng Huang.

**Software:** Jiayi Zhu.

**Supervision:** Jiayi Zhu, Kang Huang, Yuanjie Xian.

**Validation:** Yuanjie Xian.

**Visualization:** Jiyou Peng.

**Writing – original draft:** Qizheng Huang.

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
