## [Decision Letter · Decision Letter 0]

26 Dec 2024

PONE-D-24-52606Music Score Copyright Protection based on Mixed Low-order quaternion Franklin MomentsPLOS ONE

Dear Dr. Zhu,

Thank you for submitting your manuscript to PLOS ONE. After careful consideration, we feel that it has merit but does not fully meet PLOS ONE’s publication criteria as it currently stands. Therefore, we invite you to submit a revised version of the manuscript that addresses the points raised during the review process.

We look forward to receiving your revised manuscript.

Kind regards,

Sadiq H. Abdulhussain, Ph.D.

Academic Editor

PLOS ONE

Journal requirements: When submitting your revision, we need you to address these additional requirements. 1. Please ensure that your manuscript meets PLOS ONE's style requirements, including those for file naming. The PLOS ONE style templates can be found at https://journals.plos.org/plosone/s/file?id=wjVg/PLOSOne_formatting_sample_main_body.pdf and https://journals.plos.org/plosone/s/file?id=ba62/PLOSOne_formatting_sample_title_authors_affiliations.pdf. 2. Please update your submission to use the PLOS LaTeX template. The template and more information on our requirements for LaTeX submissions can be found at http://journals.plos.org/plosone/s/latex. 3. Please note that PLOS ONE has specific guidelines on code sharing for submissions in which author-generated code underpins the findings in the manuscript. In these cases, we expect all author-generated code to be made available without restrictions upon publication of the work. Please review our guidelines at https://journals.plos.org/plosone/s/materials-and-software-sharing#loc-sharing-code and ensure that your code is shared in a way that follows best practice and facilitates reproducibility and reuse. 4. We noticed you have some minor occurrence of overlapping text with the following previous publication(s), which needs to be addressed: https://www.sciencedirect.com/science/article/abs/pii/S1051200424003816?via%3Dihub In your revision ensure you cite all your sources (including your own works), and quote or rephrase any duplicated text outside the methods section. Further consideration is dependent on these concerns being addressed. 5. We note that your Data Availability Statement is currently as follows: [All relevant data are within the manuscript and its Supporting Information files.] Please confirm at this time whether or not your submission contains all raw data required to replicate the results of your study. Authors must share the “minimal data set” for their submission. PLOS defines the minimal data set to consist of the data required to replicate all study findings reported in the article, as well as related metadata and methods (https://journals.plos.org/plosone/s/data-availability#loc-minimal-data-set-definition). For example, authors should submit the following data: - The values behind the means, standard deviations and other measures reported;- The values used to build graphs;- The points extracted from images for analysis. Authors do not need to submit their entire data set if only a portion of the data was used in the reported study. If your submission does not contain these data, please either upload them as Supporting Information files or deposit them to a stable, public repository and provide us with the relevant URLs, DOIs, or accession numbers. For a list of recommended repositories, please see https://journals.plos.org/plosone/s/recommended-repositories. If there are ethical or legal restrictions on sharing a de-identified data set, please explain them in detail (e.g., data contain potentially sensitive information, data are owned by a third-party organization, etc.) and who has imposed them (e.g., an ethics committee). Please also provide contact information for a data access committee, ethics committee, or other institutional body to which data requests may be sent. If data are owned by a third party, please indicate how others may request data access.

Reviewers' comments:

Reviewer's Responses to Questions

**Comments to the Author**

1. Is the manuscript technically sound, and do the data support the conclusions?

Reviewer #1: Partly

Reviewer #2: Yes

2. Has the statistical analysis been performed appropriately and rigorously? 

Reviewer #1: I Don't Know

Reviewer #2: Yes

3. Have the authors made all data underlying the findings in their manuscript fully available?

Reviewer #1: Yes

Reviewer #2: Yes

4. Is the manuscript presented in an intelligible fashion and written in standard English?

Reviewer #1: No

Reviewer #2: Yes

5. Review Comments to the Author

Reviewer #1: Title: “Music Score Copyright Protection based on Mixed Low-order quaternion Franklin Moments”

This article takes music copyrights as an example and designs a copyright protection method for musical score digital images. To enhance the robustness of the algorithm, a new zero-watermarking method based on mixed low-order Franklin moments is proposed. The work is good and well presented. However, there are some major points need to be addressed which are as follows:

1-In general, the Abstract needs to specify the specific problem statements should be mentioned clearly to show the main contribution of this work. Besides, how the robustness has been increased using the new zero-watermarking method based on mixed low-order Franklin moments?

In addition, based on the experimental results, how the proposed algorithm has strong resistance to mixed attacks? What measures have been used and what is the rate of improvement that has been achieved?

2- What is the difference between discrete and continuous moments in QTFM? Which one is preferable to use in color image processing? Why?

3- The introduction section is weak and should be improved, more research should be added in this area. In addition, why does the ability of zero-watermarking algorithms based on orthogonal moments to resist geometric attacks remain limited? In addition, it is required to include some applications of orthogonal polynomials to show their importance such as:

i) "A steganography based on orthogonal moments." Proceedings of the international conference on information and communication technology. 2019.

ii) "Face recognition algorithm based on fast computation of orthogonal moments." Mathematics 10.15 (2022): 2721.

iii) "3D object recognition using fast overlapped block processing technique." Sensors 22.23 (2022): 9209.

Moreover, it is required to mention the main gaps that are handled in the proposed work. please list the main contributions of this work clearly.

4- It is recommended to add the section of the related works to provide a clear vision of the proposed work and its actual contribution in the context of other recent related papers.

5- There are many grammatical and typo errors that should be checked in the entire manuscript.

6- Please cite any specific figures, dataset, equation, and measurements (such as SSIM and PSNR) with reliable sources unless they are related to the authors.

7- The mathematical formula of the time complexity of recursive and non-recursive orthogonalization process are given in section 2.2. However, it is not mentioned for the proposed one. Please clarify.

Besides, it is preferable to put Fig 2 with the section of the Experimental results and discussions.

8- How Gaussian and wavelet integration methods provide an accurate computation of GKS-QFFM. Please give more explanations.

9- In general, the figures and results need more discussion with more explanation about each Tables.

10- The conclusion section is good, but it needs more improvement. It should restart the problem statement, summarize the key findings, and demonstrate the significance of the current work.

Reviewer #2: This paper addresses the challenge of designing a copyright protection method for musical score digital images. To this end, a zero-watermarking approach based on mixed low-order Franklin moments is proposed. The paper uses an invariant moments against image attacks and the consistency of global image features. The paper is well written and well organized. However, I have the following points that need to be addressed.

1. The author use some incomplete senescence. For example, the author mentioned Experimental results demonstrate that the algorithm has strong resistance to mixed attacks. The author need to mentioned the which algorithm.

2. The paper contributions need to be highlighted in the introduction. I would suggest using bullet points to make it clear to the reader.

3. Spacing should be exist between the text and reference. Please consider this point. For example � ng[12].

4. After eq 1 the author mentioned By Sn denotes. I would suggest using where Sn denotes

5. After each section, at the beginning of each section, the author needs to mention what this sections stands for. What are the contents of the section. It would be more clear to reader. Please address this issue.

6. The authors should tell the reader about the computational overhead and latency introduced by the proposed method. Just clarification is needed.

7. What are the imitations of the proposed approach. Could you please clarify this.

8. The author use ‘’Where’’ after equation 4. It should be where small letter and no spacing at the beginning of the line.

9. Dot should be placed after fig 8 caption.

10. The author should state a future work and what can be carried out in the future. This would be really helpful and open the potential for future work.

11. Some relevant references are missing. For example [R1] and [R2]

[R1] Mahmmod, Basheera M., Sadiq H. Abdulhussain, Tomáš Suk, Muntadher Alsabah, and Abir Hussain. "Accelerated and improved stabilization for high order moments of Racah polynomials." IEEE Access (2023).

[R2] Abdulhussain, Sadiq H., Basheera M. Mahmmod, Marwah Abdulrazzaq Naser, Muntadher Qasim Alsabah, Roslizah Ali, and S. A. R. Al-Haddad. "A robust handwritten numeral recognition using hybrid orthogonal polynomials and moments." Sensors 21, no. 6 (2021): 1999.

6. PLOS authors have the option to publish the peer review history of their article (what does this mean?). If published, this will include your full peer review and any attached files.

Reviewer #1: No

Reviewer #2: No

---

## [Author Response · Author response to Decision Letter 1]

25 Feb 2025

We are very grateful to reviewers’ critical comments and thoughtful suggestions. Based on these comments and suggestions, we have made careful modification on the original manuscript. All changes made to the text are in red in the revised manuscript so that they may be easily identified. Some of your questions were answered in response letter.

Once again, we acknowledge your comments and constructive suggestions very much, which are valuable in improving the quality of our manuscript.

---

## [Decision Letter · Decision Letter 1]

9 Apr 2025

Music Score Copyright Protection based on Mixed Low-order quaternion Franklin Moments

PONE-D-24-52606R1

Dear Dr. Zhu,

We’re pleased to inform you that your manuscript has been judged scientifically suitable for publication and will be formally accepted for publication once it meets all outstanding technical requirements.

Kind regards,

Sadiq H. Abdulhussain, Ph.D.

Academic Editor

PLOS ONE

Additional Editor Comments (optional):

Reviewers' comments:

Reviewer's Responses to Questions

**Comments to the Author**

1. If the authors have adequately addressed your comments raised in a previous round of review and you feel that this manuscript is now acceptable for publication, you may indicate that here to bypass the “Comments to the Author” section, enter your conflict of interest statement in the “Confidential to Editor” section, and submit your "Accept" recommendation.

Reviewer #1: All comments have been addressed

Reviewer #2: All comments have been addressed

2. Is the manuscript technically sound, and do the data support the conclusions?

Reviewer #1: Yes

Reviewer #2: Yes

3. Has the statistical analysis been performed appropriately and rigorously? 

Reviewer #1: I Don't Know

Reviewer #2: Yes

4. Have the authors made all data underlying the findings in their manuscript fully available?

Reviewer #1: Yes

Reviewer #2: Yes

5. Is the manuscript presented in an intelligible fashion and written in standard English?

Reviewer #1: Yes

Reviewer #2: Yes

6. Review Comments to the Author

Reviewer #1: Most of the comments in the revised version have been properly addressed and no further comments are needed.

Reviewer #2: The author addressed all the issues. The paper is now ready for consireation to be published. The currect format is fine I would recommand accepting this paper.

7. PLOS authors have the option to publish the peer review history of their article (what does this mean?). If published, this will include your full peer review and any attached files.

Reviewer #1: No

Reviewer #2: No

---

## [Editor Report · Acceptance letter]

PONE-D-24-52606R1

PLOS ONE

Dear Dr. Zhu,

I'm pleased to inform you that your manuscript has been deemed suitable for publication in PLOS ONE. Congratulations! Your manuscript is now being handed over to our production team.

Kind regards,

on behalf of

Dr. Sadiq H. Abdulhussain

Academic Editor

PLOS ONE